The physiological consequences of varied heat exposure events in adult Myzus persicae: a single prolonged exposure compared to repeated shorter exposures

Ghaedi Behnaz ghaedi.b@gmail.com bghaedi@myune.edu.au
Andrew Nigel R.
Centre for Behavioural and Physiological Ecology, Zoology, University of New England , Armidale , NSW , Australia
Chrostek Ewa
Electronic publication date: 2016 Aug 2
Publication date: 2016
Volume: 4
Electronic Location ID: e2290
Received 2015 Dec 30; Accepted 2016 Jul 5
Copyright: ©2016 Ghaedi and Andrew
Copyright year: 2016
Copyright holder: Ghaedi and Andrew
License: This is an open access article distributed under the terms of the Creative Commons Attribution License, which permits unrestricted use, distribution, reproduction and adaptation in any medium and for any purpose provided that it is properly attributed. For attribution, the original author(s), title, publication source (PeerJ) and either DOI or URL of the article must be cited.
License URL: https://creativecommons.org/licenses/by/4.0/

Keywords: Fluctuation temperature, Metabolite, Aphid, Heat exposure, Repeated heating, Thermal tolerance

Funding: UNE Funds were provided by UNE as a scholarship to Behnaz Ghaedi as well as for equipment and consumables used in the study. The funders had no role in study design, data collection and analysis, decision to publish, or preparation of the manuscript.

==============================
The study of environmental stress tolerance in aphids has primarily been at low temperatures. In these cases, and in the rare cases of high temperature tolerance assessments, all exposures had been during a single stress event. In the present study, we examined the physiological consequences of repeated high temperature exposure with recovery periods between these stress events in Myzus persicae. We subjected individuals to either a single prolonged three hour heating event, or three one hour heating events with a recovery time of 24 h between bouts. Aphids exposed to repeated bouts of high temperatures had more glucose and higher expression of proteins and osmolyte compounds, such as glycerol, compared to the prolonged exposure group. However, aphids exposed to the repeated high temperature treatment had reduced sources of energy such as trehalose and triglyceride compounds than the prolonged exposure group. Recovery time had more physiological costs (based on production of more protein and consumption of more trehalose and triglyceride) and benefits (based on production of more osmolytes) in repeated high temperature treatments. As aphids are known to respond differently to constant versus ‘natural’ fluctuating temperature regimes, conclusions drawn from constant temperature data sets may be problematic. We suggest future experiments assessing insect responses to thermal stress incorporate a repeated stress and recovery pattern into their methodologies.

Introduction

Climate change is one of the most critical threats to biodiversity (Dawson et al., 2011; IPCC, 2014). Human-induced climate change is predicted to increase the frequency of climatic extremes (e.g., heat waves and severe droughts or floods), climatic variability (Lean & Rind, 2008; Rahmstorf & Coumou, 2011) and mean of the thermal environment in regions around the world (Niehaus et al., 2012). Global mean temperature has risen by 0.85 °C from 1880 to 2012. All of the warmest 20 years on record have occurred since 1990 (CSIRO-ABM, 2014).

Exposure to different thermal means and variability can have a substantive influence on animal physiological performance (Huey et al., 2012). This may include a modification of performance curve parameters including two critical fitness-influencing components: the upper critical temperature and the thermal optimum (Angilletta, 2009). All insects keep their physiological performance within a specific range of temperatures, and many of their physiological functions may be reduced when exposed to extreme temperatures (Mironidis & Savopoulou-Soultani, 2010). An important aspect of habitat quality is minimal exposure to extreme thermal stress (Huey, 1991). In terrestrial habitats, climate change is strengthening fluctuations and amplitude of temperature variation, which will lead to prolonged adverse temperature exposure in terresterial habitats (Sinclair et al., 2006; Andrew, 2013; Vasseur et al., 2014).

Critical temperature responses can shift somewhat based on an animals environmental thermal experience (Somero, 2010). As insects are ectotherms, their survival, population dynamics, and distribution are influenced by temperature (Chown et al., 2010; Bauerfeind & Fischer, 2014). Therefore, investigating impacts of temperature variation on an individual’s performance and understanding these plastic responses of populations to temperature is critical (Pörtner et al., 2006); and may have implications for their potential responses to climate change.

Some aphid species, including M. persicae, are distributed widely across the globe. The cabbage aphid, Brevicoryne brassicae (L.), the turnip aphid, Lipaphis erysimi Kalt, and the green peach aphid, Myzus persicae (Sulzer) (Hemiptera: Aphidae) are the three major aphid pests infesting canola in Australia (Gu, Fitt & Baker, 2007; Gia & Andrew, 2015). Myzus persicae (green peach aphid) is known to transmit over 100 phytopathogenic viruses among 50 different plant families. Many of its hosts include major crops (e.g., sugar beet, beans, brassicas, potatoes, citrus) and on a world wide scale this species is regarded as the most important aphid pest (Blackman & Eastop, 1984). Green peach aphid host alternating between the primary peach Prunus persica host in winter and various herbaceous hosts, belonging to 50 different families, which include brassicas, potatoes and sugar beet, in summer (Blackman & Eastop, 2000).

Extreme climatic events such as heat waves and daily fluctuations can impact aphids in various ways: reducing fecundity and population growth; slowing development; and can also affect community structure and interrupt trophic cascades through effects on performance of individual species and through changes in the strength of interactions between them (Davis, Radcliffe & Ragsdale, 2006; Gillespie et al., 2012; Zhao et al., 2014; Colinet et al., 2015). Aphids are more sensitive to acute changes in temperature rather than the duration of extreme temperature exposure, and nymphs do not experience diapause (Davis, Radcliffe & Ragsdale, 2006), making them a suitable test organism for our study. Studying the effects of climate change on aphids is complex. Many organisms that live in fluctuating thermal environments, such as aphids, display a high degree of thermal plasticity in their response to changing conditions, and have a greater resilience in their ability to recover from ecological perturbations (Davis, Radcliffe & Ragsdale, 2006). In order to understand how aphids perform in a warming climate, it is necessary to consider both exposure temperature and exposure duration with or without recovery time.

In responding to extreme environmental conditions, insects rely on a combination of different factors: such as molecular processes (gene expression, heat shock proteins, and enzymes); changes in membrane structure; and osmolyte compounds; to survive and recover from unfavourable conditions (Back, Oakenfull & Smith, 1979; Henle, Peck & Higashikubo, 1983; Lin, Hefter & Ho, 1984; Kim & Lee, 1993; Meng et al., 2004; Wang et al., 2006; Clark & Worland, 2008; Tollarová-Borovanská, Lalouette & Koštál, 2009). Aphids and whiteflies were one of the first insect taxa reported to accumulate polyols in response to high temperatures and also reveal that these compounds could stabilize proteins structure against thermal denaturation (Back, Oakenfull & Smith, 1979). Whole-animal respiration will change when exposed to repeated stress (Lalouette et al., 2011; Yocum et al., 2011) and understanding the implications of this is critical as this will impact on animal survival and body maintenance as changes occur in energy and water usage (Schimpf et al., 2009; Chown, Sorensen & Terblanche, 2011). In addition, the upper critical temperature threshold at which an animal loses muscular control (CTmax) is a metric of interest as it is useful in predicting phenotypic effects of warming (Huey et al., 2012). Using laboratory estimates of thermal tolerance enables the seasonal abundance and geographic distribution of organisms to be determined (Ju et al., 2013).

It has become clear that constant temperatures are not useful for study performance thermal response of organisms as they do not mimic the fluctuating temperature conditions which occur within the natural environment (Lamb, 1961; Davis, Radcliffe & Ragsdale, 2006). Constant temperature conditions also underestimate thermal thresholds of individuals and become less accurate compared to fluctuating temperature regimes (Davis, Radcliffe & Ragsdale, 2006; Niehaus et al., 2012). This implies that fluctuating temperature experiments ought to be “normal” while constant temperature insect development studies were “abnormal” experimental conditions (Cloudsley-Thompson, 1953). Fluctuating temperatures enhanced resistance of leaf beetles and fruit-flies to low temperature (Casagrande & Haynes, 1976; Meats, 1976) and that fitness could be more significant in repeated temperatures (Beardmore & Levine, 1963).

In previous studies much attention has focused on the cold tolerance of Myzus persicae (O’doherty & Bale, 1985; Bale, Harrington & Clough, 1988; Clough, Bale & Harrington, 1990; Howling, Bale & Harrington, 1994; Bezemer, Jones & Knight, 1998; Vorburger, 2004) as well as assessments of the effects of temperature on physiological parameters including their thermal tolerance at different latitudes and altitudes (Bezemer, Jones & Knight, 1998; Alford, Blackburn & Bale, 2012a). There is a paucity of knowledge regarding how aphid physiology is affected by repeated exposure to high temperatures. Here, we assessed aphid metabolic rates after different heating regimes (a single prolonged exposure and repeated short-term exposure with recovery time, and no heat exposure) using flow-through respirometry; we also assessed the upper critical temperature threshold, energy reserves and their osmolyte compound profile, as indicators of aphid stress response after exposure to the different heating regimes.

We predict that repeated short-term exposure to high temperatures will increase adult aphid thermal tolerance, as recovery time between heat stress bouts will enable metabolic and cellular repairs to occur (Storey & Storey, 2004). We also predict that aphids will accumulate polyols and sugars in response to high temperatures (Hendrix & Salvucci, 1998) which will result in an increased CTmax.

Figure 1 Diet cages.

These were constructed from rigid clear PVC plastic tubes, 12 cm in diameter (aphid feeding) and covered with a 7 × 7 cm sheet of Parafilm laboratory film.

Methods and Materials

Aphids and diet chamber construction

Stock colonies of Myzus persicae (green peach aphid), were collected from canola plants at the Laureldale Farm property (University of New England, Armidale, NSW Australia) in 2013.

The lab colony were established from an isofemale line (BG_13-001) and maintained on canola variety ‘Thunder Bay’ plants in a glasshouse with 25 ± 0.5 °C temperatures, 65% relative humidity, under a 16:8 h (L:D) photoperiod provided by fluorescent lamp for two years to ensured continuous apomictic parthenogenesis.

Since aphids have parthenogenetic embryogenesis (Dixon, 2005) we bred three generations of aphids in the lab to reduce maternal and grand maternal affects: we removed the wingless female adult aphids after 3 days, then the resulting nymphs were left for seven days and we continued this process for three generations. The third generation resulting nymphs were housed at 25 °C, 65% relative humidity, under a 16:8 h (L:D) in a Thermoline incubator (TRH-300) prior to experimental manipulation.

Diet cage preparation

To rear our aphids on artificial diets we used diet cages. Diet cages were constructed from rigid clear PVC plastic tubing with a 12 cm diameter (Fig. 1). A 7 cm × 7 cm square of parafilm was extended by force across the top of a cage, and 2 ml of artificial diet was pipetted on top of the parafilm layer. A second 7 cm × 7 cm sheet of parafilm was then extended over the first sheet, forcing diet across the top surface of the cage, but avoiding leakage over the edge. Then by using a paintbrush, aphids were placed on the underside of the parafilm. The diet cages where then put into a closed transparent box (30 × 30 cm) and put into the incubator prior to the application of treatments.

Artificial diet

The artificial diet used to rear the aphids was based on work by Douglas (1988) and consisted of specific concentrations of phosphate, vitamins, and minerals (Table 1). In the experiments, four replicate groups of 200 larval aphids were maintained on the test diet for 5 days (i.e., from one to 6 days of age). The specified amounts of each component (see Table 1) were prepared with distilled deionized water to the total volume of 10 ml in a glass container. The pH of the solution was 7.0–7.5. The diet solution was divided into 2 ml aliquots, and stored at 4 °C s for less than 1 week or at −20 °C for less than three months. In all experiments, distilled deionized water was used in all of the solutions and the diets of the aphids were changed twice a week for the duration of our study.

Table 1 Concentrations and composition of the artificial diet used to rear Myzus persicae.

Molecular weight (MW); molar Mass (mM) and mole percent (Mol%).

Amino acid	MW	Mol %	150 mM	Sucrose mix	1,000 mM	
Alanine	89.09	3.8	50.8 mg	Ascorbic acid	10 mg	
Asparagine	150.1	9.5	213.9 mg	Citric acid	1 mg	
Aspartate/Aspartic acid	133.1	9.5	189.7 mg	MgSO4	11 mg	
Cysteine	157.6	1.8	42.5 mg	Sucrose	3,400 mg	
Glutamic acid	147.13	5.6	123.6 mg	Mineral stock		
Glutamine	146.1	11	241.1 mg	FeCl3	13.1 mg	
Glycine	75.07	0.8	9.0 mg	CuCl2.2H2O	1.7 mg	
Proline	115.1	3.8	65.6 mg	MnCl2.4H20	4 mg	
Serine	105.09	3.8	59.9 mg	ZnCl2	13.6 mg	
Tyrosine	181.2	0.4	10.9 mg	Vitamin stock		
Arginine	210.66	9.5	300.2 mg	Biotin	0.1 mg	
Histidine	209.6	5.8	182.4 mg	Pantothenate	5 mg	
Isoleucine	131.18	5.8	114.1 mg	Folic acid	2 mg	
Leucine	131.18	5.8	114.1 mg	Nicotinic acid	10 mg	
Lysine	182.6	5.8	158.9 mg	Pyridoxine	2.5 mg	
Methionine	149.2	1.9	42.5 mg	Thiamine	2.5 mg	
Phenylalanine	165.2	1.9	47.1 mg	Choline	50 mg	
Threonine	119.1	5.8	103.6 mg	Myo-inositol	50 mg	
Tryptophan	204.2	1.9	58.2 mg	Phosphate		
Valine	117.1	5.8	101.9 mg	K2PO4.3H20	150 mg	

Experimental conditions

To investigate the effects of fluctuating thermal regimes (FTR), the experimental design was a simplified version of the experimental protocol of Marshall & Sinclair (2011) and Zhang et al. (2011). For the repeated heat stress exposure treatment, adult aphids were exposed to one to three diurnal cycles (We left them in the food diet and put them into the incubator) of 1h at 38 °C (we didn’t control RH) followed by 24 h at 25 °C, 65% relative humidity (RH). Adult aphids (seven day old individuals) from the third generation reared on the artificial diet were used. For the single prolonged heat exposure treatment, separate groups of adults were exposed to 38 °C for 3 h, to make total heat exposure time equivalent among treatments. Post treatment, this group were also given a 24 h-period of recovery at 25 °C, 65% RH. Meanwhile, control animals were held at 25 °C, 65% RH for the duration of the study.

Estimates for the upper critical temperature threshold limits for M. persicae have previously found to range from 38.5 °C (Broadbent & Hollings, 1951) to 42 °C (Hazell et al., 2010). Myzus persicae are known to survive 1h per day above their CTmax of 38.5 °C (Davis, Radcliffe & Ragsdale, 2006) and so the FTR was chosen to fluctuate around a value close to their CTmax. In general, upper temperatures are difficult to experiment with compared to lower temperatures because their performance curve optima is very close to CTmax. A test temperature of 38 °C was chosen as there was 60–70% survival following the prolonged thermal exposure for 3 h.

Quantification of metabolic reserves

Twenty-four hours after a heat exposure (Fig. 2), four replicate groups of adult aphids were weighed to 0.001 mg on a Mettler Toledo XP2U (Switzerland) electronic balance, then homogenized in 80 µl extraction buffer (35 mM Tris, 25 mM KCl, 10 mM MgCl2, pH 7.5) with 0.1% (v/v) Triton-X-100 after centrifuging for 1 min at 13,000 rpm at 4 °C. Supernatant was then removed and placed in a new tube which was then stored at −20 °C until all assays were conducted.

The assay kits: used included the Sigma triglyceride and glycerol assay kit (Sigma Triglyceride (Sigma T2449; Sigma-Aldrich), Free glycerol reagent (Sigma F 6428; Sigma-Aldrich), Glycerol stock (Sigma G7793-5 ml at 2.5 mg/ml)); the BioRad Coomassie Brilliant Blue microassay method (500-0201), with bovine serum albumin as standard (40–480 mg protein) for protein; and the Sigma glucose assay kit for glucose (Product Code GAGO-20, contains 500 units of glucose oxidase, O-Dianisidine Reagent (Product Code D 2679)) and following trehalose measurement (Porcine Kidney Trehalase (Sigma T8778; Sigma-Aldrich) and 0.2 M sodium citrate (5.882 g/100 ml), 1 mM Sodium EDTA (37.22 mg/100 ml), D + Trehalose dihydrate (Sigma T0167; Sigma-Aldrich)).

Figure 2 Experimental design.

All experiments were performed on adult aphids, that were 6 days old on the first experimental day. Each black rectangle represents a 1 h exposure to 38 °C. The blue rectangle represents a 3 h exposure to 38 °C. Adults were kept at 25 °C as the control group (yellow line). Red dots indicate sampling points. All samples were collected 24 h after final treatments.

For these assays, we measured Glucose, Triglyceride, Glycerol, Trehalose and total protein content in triplicate and we used a visible wavelength spectrophotometer (Epoch Microplate Spectrophotometer; BioTek, Winooski, VT, USA) with absorbance at 544, 540 and 750 nm and 96 well plates, with volumes scaled down from the manufacturer protocols (Ridley et al., 2012).

Active metabolic rate measurement (AMR)

Flow-through CO2-based respirometry was used to record VCO2, with a similar experimental setup as described by Terblanche & Chown (2007). A HiBlow HP40 air pump was used to feed atmospheric air into sodalime (VWR with indicator AnalaR NORMAPUR analytical reagent) and Drierite (WA Hammond Drierite Company) scrubber columns, to remove CO2 and water vapour from the air stream, respectively. The flow rate of the airstream was regulated at 80 ml min−1 by a flow control valve (Model 840, Sierra Side-Trak; Sierra Instruments Inc., Monterey, CA, USA), connected to a mass flow controller (Sable MFC-2). Thereafter, air flowed through the zero channel (cell A) of a calibrated (to 6 ppm CO2 in air) infrared CO2–H2O analyzer (Li-7000; Li-Cor, Lincoln, NE, USA). The airstream then flowed over the test animal in the 5ml glass cuvette, which was placed in a programmable water bath (Grant, GP200-R4), programmed using LABWISE software with increasing temperature of 0.25 °C min−1 (see Basson & Terblanche, 2010). The air leaving the cuvette then entered the analyser through another channel which recorded the difference in CO2 concentration of the air before and after it flowed through the cuvette, at 1 s intervals. Changes in animal position (activity) were recorded using an infrared activity detector (AD-1; Sable Systems, Las Vegas, NV, USA). Aluminium foil was placed around this cuvette to restrict light exposure and to ensure high quality activity recordings (MacMillan et al., 2012).

Four aphids per replicate were weighed to 0.001 mg on an electronic microbalance before and after the experiment and mean mass used as a covariate in statistical analyses. The output from the CO2 analyzer and activity data were recorded with the LiCor 7000 software and analysed using Expedata V1.25 software (Sable Systems International, Las Vegas, NV, USA). Volumes of CO2 in ppm were corrected for baseline drift and then converted to µl CO2 h−1 using Expedata software. Rates of CO2 production (in µl CO2 h−1) were calculated from the whole record by transforming ppm concentration of CO2 to CO2 fraction and then multiplying by the flow rate (80 ml min−1). The area under the curve (integral of ml CO2 min−1 vs min) was calculated. This area was equal to the volume of CO2 produced by each replicate in the cuvette, and this volume was divided by the total period of measurement (2.30 h), multiplied by 1,000 to give µl CO2 h−1 to give the metabolic rate per aphid per hour (Castañeda et al., 2009). Metabolic rate was measured for each of the four times two replicate aphids after the final exposure in all experimental groups.

Measuring upper critical thermal limits (CTmax)

Critical thermal limits were determined by subjecting aphids to a regime of increasing temperatures and monitoring their ability to control their movement: loss of muscular controls to determine their CTmax threshold. Forty adult aphids were each placed within an individual 5 ml plastic tube at a pre-set temperature (25 °C). A programmable water bath was set to increase the temperature from 25 °C to 35 °C with a rate of 0.5 °C min−1 then the temperature was increased to 45 °C with a rate of 0.1 °C min−1 to minimize the hardening response across a broad range of upper critical temperatures and to determine the temperature at which walking ceased and aphids succumbed (Hazell et al., 2008; Alford, Blackburn & Bale, 2012b). Upper critical temperature was measured for 40 adult aphids for each of the three experimental treatments after final heat exposure in all experiments group.

Statistical analysis

All data were analysed for normality and tested for homogeneity of variances for treatment means using the Levene’s Test, in the HOVTEST option of GLM procedure within SAS software (2008). A completely randomized design was employed. One-way analysis of variance was performed using the GLM procedure. Tukey post hoc tests were used to compare means (P < 0.05).

Figure 3 Mean (±s.e) (A) Glucose, (B) Trehalose, (C) Protein, (D) glycerol and (E) Triglyceride content of Myzus persicae during repeated (R1, R2, R3), prolonged (P) and control state (C1, C2, C3).

The metabolite content is based on the mean of four replicates, two aphids per replicate for the all metabolites. Post-hoc pairwise differences among treatments indicated by stars.

Table 2 Energy reserves, osmolytes concentration and protein mass of Myzus persicae in three treatment groups: maintained at 25 °C (control), heated for a single bout of 3 h at 38 °C (1 × 3 h) and heated for three bouts of 1 h at 38 °C (3 × 1 h).

Significant values in bold.

Treatments	Treatments	Contrasts	
Measurements				C3–R3	C3–P	P–R3	C3–R3–P	
	D.F	F Value	P-value	D.F	F Value	P-Value	D.F	F Value	P-Value	D.F	F Value	P-Value	D.F	F Value	P-Value	
Glucose	6	22.31	<0.0001	1	39.71	<0.0001	1	25.85	<0.0001	1	1.48	0.2262	2	43.22	<0.0001	
Trehalose	6	4.40	0.0005	1	21.06	<0.0001	1	0.96	0.3297	1	13.03	0.0005	2	10.34	0.0017	
Protein	6	15.85	<0.0001	1	84.09	<0.0001	1	32.54	0.3719	1	12.01	<0.0001	2	73.75	0.0004	
Glycerol	6	23.24	<0.0001	1	67.76	<0.0001	1	16.08	0.0001	1	17.82	<0.0001	2	49.96	<0.0001	
Triglyceride	6	3.57	0.0029	1	16.02	0.0001	1	0.55	0.4614	1	10.65	0.0015	2	7.49	0.0073	
Measurements	Contrasts	
	C3–P–R3	C3–P	C3–R3	P–R3	
	D.F	F-Value	P-Value	D.F	F-Value	P-Value	D.F	F-Value	P-Value	D.F	F-Value	P-Value	
CO2	2	0.48	0. 62	1	0.07	0.967	1	4.65	0.481	1	3.55	0.312	
H2O	2	3.61	0.04	1	11.24	0.012	1	17.69	0.005	1	0.73	0.872	
CTmax	2	1.63	0.127	1	3.61	0.759	1	0.09	0.059	1	2.54	0.113	

Results

Glucose content

The glucose content of the control group exhibited no significant differences over the course of the experiment (Day 1: 1.61 ± 0.02 to Day 3: 1.96 ± 0.06 µg glucose mg−1 (Fig. 3A)). However, adult aphids in both the prolonged and repeated exposure groups experienced a significant increase in glucose content. After a single cycle of 38 °C for 1h and 25 °C for 24 h, glucose content increased significantly, nearly one and half fold for aphids in the repeated exposure group (P < 0.0001; Table 2). For the duration of the experiment, the glucose content of aphids in the repeated exposure group peaked at 3.26 ± 0.02 µg glucose mg−1 (Fig. 3A). For adult aphids continuously exposed to 38 °C for 3 h (prolonged exposure treatment), glucose content significantly increased compared to the control group (3.01 ± 0.02 µg glucose mg−1).

Trehalose content

The trehalose content of aphids exposed to repeated heating events significantly decreased from 0.44 ± 0.01 to 0.27 ± 0.01 µg trehalose mg−1 after the second heating exposure (Fig. 3B). After three cycles of heating, trehalose content of aphids was significantly lower than the control group on the third day of the experiment (P < 0.001; Table 2). For aphids exposed to 38 °C for 3 h, the trehalose content did not significantly differ (P = 0.329; Table 2) from that of the control group.

Protein content

Similar to glucose and glycerol content, protein content of aphids that were heated during repeated exposures was significantly higher than in the control group and their prolonged exposure counterparts (P < 0.001; Fig. 3C). After one cycle of 1 h at 38 °C and 24 h at 25 °C, the protein content of aphids was 17.16 ± 0.06 µg protein mg−1, but this value steadily increased every cycle, and by the end of third cycle protein content was 19.79 ± 0.1 µg protein mg−1 (Fig. 3C). Protein content in aphids heated at 38 °C for 3 h did not significantly differ from the control (Fig. 3C).

Glycerol and triglyceride content

Triglyceride content in aphids did not significantly differ between control and the prolonged exposure groups. However there was a significant drop of triglyceride content in aphids after three cycles of repeated heating in the repeated exposure group: triglyceride content in the repeated exposure group was less than half that of control (0.659 ± 0.01) and prolonged (0.597 ± 0.01) aphids and reached 0.324 ± 0.02 µg triglyceride mg−1 (P < 0.002; Fig. 3E).

Glycerol content was significantly higher (P < 0.0001; Table 2) in aphids of the repeated exposure group increasing from 1.53 ± 0.01 to 1.96 ± 0.02 µg triglyceride mg−1 after three cycles of heating. There was also significantly higher glycerol content in aphids of the prolonged exposure group compared to the control group (P = 0.0001; Fig. 3D).

Thermal tolerance

There was no significant difference in aphid CTmax between the thermal treatments (Fig. 4).

Active metabolic rate measurements

There was no significant difference in metabolic rate 24 h after heating among treatments (P = 0.6; Fig. 5A). Aphids from the repeated heating and prolonged heating treatments exhibited significant water loss compared to the control group (P = 0.04; Fig. 5B).

Discussion

In the present study, we examined the physiological consequences of either being exposed to repeat short-period high temperatures bouts with recovery time periods between them, or a single prolonged temperature treatment, in the aphid Myzus persicae (Fig. 2). We observed increased physiological costs and benefits during repeated heating exposure: this group of aphids had more glucose and higher expression of proteins and osmolyte compounds such as glycerol compared to the prolonged exposure group. However, the repeated high temperature exposure group also had fewer sources of energy such as trehalose and triglyceride compounds compared to the prolonged exposure group. We found that recovery time had more physiological costs (based on production of more protein and consumption of more trehalose and triglyceride) and benefits (based on production of more osmolytes) for repeated high temperature exposure group, but interestingly we saw no changes in thermal tolerance and metabolic rate across treatments.

Figure 4 Mean (±s.e) thermal tolerance point (CTmax) of aphids among control (C), prolonged (P) and repeated (R) exposure treatments.

Figure 5 Mean (±s.e) (A) CO2 production (µl CO2 h−1) and (B) H2O output rates (µg H2O h−1) of aphids among control (C), prolonged (P) and repeated (R) exposure treatments.

Metabolic rate based on four times two replicate. Post-hoc pairwise differences among treatments indicated by stars.

Impacts of repeated high temperature on thermal tolerance of aphids

Fluctuating temperature studies are better at predicting the thermal tolerance of aphids than constant temperature studies, even when the mean temperatures are the same between constant and fluctuating regimes (Lamb, 1961); constant temperature studies also underestimate critical temperature thresholds as it is known that fluctuating temperatures develop threshold limits (Casagrande & Haynes, 1976; Meats, 1976). The upper critical temperatures in M. persicae start from 38.5 °C (Broadbent & Hollings, 1951) to 42 °C (Hazell et al., 2010). To our knowledge just two studies (Davis, Radcliffe & Ragsdale, 2006; Gillespie et al., 2012) have assessed fluctuating high temperatures in M. persicae: but our study is the first to assess physiology of aphids at high temperatures. Davis, Radcliffe & Ragsdale (2006) examined the effect of high and fluctuating temperatures on the development of M. persicae. They demonstrated that under fluctuating temperatures, M. persicae had greater fertility and faster development and had the capacity to survive 1 h every day above the CTmax of 38.5 °C (Davis, Radcliffe & Ragsdale, 2006). These results are in contrast to Gillespie et al. (2012) who found that under heat wave conditions, the growing population in aphids was lower when exposed to heat waves than weather with periodic hot days. Therefore, differences in their results may be due to other factors such as changes in mean temperatures.

Extreme temperature exposure in short bursts often increase survival compared to prolonged exposure. At optimum conditions, injury repair can occur by restoring ion homeostasis (Kostal et al., 2007) and replenishing energy levels. In Drosophila melanogaster, Krebs & Loeschcke (1994) found similar results for effects of heating in adult female flies by exposing them to 36 °C 1–3 times with 48 h rest between heating exposure. Their results showed that exposure to bouts of stress increased survival temperature to 39 °C.

Furthermore, recovery time at optimum temperatures may have permitted aphids to recover from the negative impacts of high temperatures (Hazell et al., 2010). In our study, it appears that recovery times between high temperatures can repair injuries, as aphids exhibited higher levels protein and osmolyte in the repeated exposure treatment compared to the prolonged exposure treatment group: but there was no significant difference in their CTmax between treatments. One possible explanation is that the proteins and osmolytes were upregulated during the recovery time, but upregulation was not great enough to change their thermal threshold between treatments (Tammariello, Rinehart & Denlinger, 1999). One further reason for conservatism of physiological resistance to heat is due to upper thermal tolerance being largely uncorrelated to estimates of natural temperature (Grigg & Buckley, 2013) as a high number of terrestrial organisms are unlikely to evolve physiological resistance to increased heat (Kellermann et al., 2012; Hoffmann et al., 2013). In such cases, development of physiological resistances will be weakened.

Changes in energy reserves at fluctuating temperatures

Acclimation impacts the structure of lipid layers (Hazel, 1995), sugar or polyol amount (Hendrix & Salvucci, 1998) and metabolic rate (Hoffmann & Parsons, 1997): all of which can influence temperature resistance (Andersen et al., 2010). Whiteflies and aphids appear to be the first reported organisms that collected polyols in response to high temperatures (Hendrix & Salvucci, 1998). Sugars and polyols balance out and stabilize the natural structure of proteins, protecting them from warm denaturation (Back, Oakenfull & Smith, 1979).

The osmolyte compounds in aphids heated continuously for 3 h were less than those aphids exposed to three 1 h cycles of repeated high temperatures. Adaptations to surviving high temperatures, such as stress protein production and the synthesis of resources like glucose and trehalose (Hottiger et al., 1994; Jain & Roy, 2009). Accordingly, we anticipated that repeated high temperatures would result in the consumption of vital energy reserves.

Our results indicate that repeated high temperature exposure is physiologically costly for aphids compared to prolonged high temperature exposure, as there is a significantly lower amount of trehalose and triglyceride production after repeated high temperatures. Trehalose and proteins play important roles in stress responses (Parsell & Lindquist, 1993; Feder & Hofmann, 1999; Salvucci, Stecher & Henneberry, 2000; Jain & Roy, 2009; Smith et al., 2012). Induction of thermal tolerance by trehalose is also inferred from the fact that the level of trehalose is correlated with thermal tolerance (Hottiger et al., 1994): with this in mind, we measured the content of these two known compounds in M. persicae. Protein content was higher in aphids of the repeated high temperature treatment group, which has a well-defined role as a protective material at high temperatures (Jain & Roy, 2009) and this result is in agreement with previous studies (Huang, Chen & Kang, 2007; Tollarová-Borovanská, Lalouette & Koštál, 2009; Zhang et al., 2011). We hypothesise that in repeated high temperature exposure, protein production is triggered once the aphids are returned to their recovery temperature: a trigger that is not available to aphids exposed to prolonged periods of high temperature.

An unusual result in our study was that the amount of trehalose decreased in aphids exposed to repeated high temperatures compared to the aphids exposed to prolonged high temperatures. This contradicts the findings of previous studies assessing trehalose after exposure to high temperatures (Hottiger et al., 1994; Jain & Roy, 2009) but does support the findings of Teets et al. (2010). Trehalose may be used to produce an osmolyte, such as mannitol, in aphids which is important at high temperatures (Hendrix & Salvucci, 1998). Trehalose can also function as a reserve carbohydrate in others, which acts as a protectant. For example, it has been noted that trehalose accumulates under conditions of environmental stress, such as desiccation and is rapidly metabolized in rehydration. A reserve function could be inferred. Trehalose can also accumulate in response to other stresses, such as heat or osmotic shock (Newman, Ring & Colaco, 1993), for example Lalouette et al. (2007) demonstrated that trehalose can be changed to glycogen therefore is related to energy storage functions.

Triacylglycerols constitute a large part of the aphid lipid assemblage (Itoyama et al., 2000) and serve as a reservoir for fatty acids that can be used for energy production. Very large amounts of triacylglycerol can occur in aphids, comprising 20–30% of fresh body weight (Strong, 1963; Sutherland, 1968). According to expectations, we observed lower amount of triglyceride content after repeated high temperatures compared to the prolonged treatment aphids. One explanation is, these storage lipids are used as a source of metabolite energy for physiological processes: many insects use lipids as an energy source, and fatty acids are stored in the fat body in the form of triacylglycerol (Blacklock & Ryan, 1994; Itoyama et al., 2000). Since glycerol is known to maintain cells from hyperthermic cell death, induced thermal protection and warming protection may be applied by adjustment of either protein or membranes (Henle & Warters, 1982). Our results demonstrated that glycerol in aphids exposed to repeated extreme heat exposure (by the end of the third cycle) is higher than the levels in aphids exposed to prolonged thermal extremes; and this is in accordance with the role of this compound during heat stress (Benoit et al., 2007).

Aphids are very sensitive to acute changes in temperature rather than the duration of stress (Davis, Radcliffe & Ragsdale, 2006). We found significant changes in metabolite reserves after repeated high temperature exposure compared to the prolonged temperature exposure treatments with the same duration indicating that the number of extreme high temperature events is more important than the duration of extreme temperatures although the CTmax of aphids was constant for both temperature regimes.

Impacts of repeated high temperature on metabolic rate of aphids

Respiration is the first process restricted in animals at low and high temperatures, and is connected to the limitation of blood circulation and ventilation (Portner, 2001). We observed no significant differences in metabolic rate 24 h after heating and this is in contrast to previous studies looking at metabolic rates at upper critical temperature (Klok, Sinclair & Chown, 2004; Boardman, Sorensen & Terblanche, 2013). Elevated standard metabolic rate after stressful conditions is a repair cost (Boardman, Sorensen & Terblanche, 2013). We measured active metabolic rate with increasing temperatures not resting metabolic rate or standard metabolic rate at 25 °C. In addition, we also found that there was a lower rate of water loss after heating: this may be due to aphids closing their spiracles to reduce water loss after heating events, and possibly a strategy for heat resistance.

Outlook

To date, the key assessment of environmental stress tolerance in the aphid, M. persicae has been at low temperatures and in most known studies; the aphids have been exposed to a single stress event. In natural conditions, M. persicae is exposed to repeated bouts of high temperature punctuated by periods of recovery. Furthermore, because aphids are known to respond differently to consistent versus more ‘normal’ fluctuating temperatures, conclusions drawn from constant temperature studies may be unreliable. We suggest future experiments incorporate a repeated stress and recovery pattern into their methodologies to fully assess responses to extreme temperature exposure to reflect the natural growing conditions of this species of aphid and also, incorporate biological and fitness studies to physiological studies at the same time to reach comprehensive achievements in this filed.

We thank Emma Ridley for her suggestions in my nutritional assays, Ary Anthony Hoffmann, John Terblanche, Jesper Givskov Sørensen, Katie Marshall and Leigh Boardman for their suggestions in experimental design development and Berlizé Groenewald for assistance with respirometry setup and calculations and Hamid Reza Hemati Matin for his assistance in analysing data. We thank Bianca Boss-Bishop and Sarah Hill for commenting on previous version of the manuscript and Leigh Boardman and two anonymous referees for suggestions that led to an improved manuscript.

Additional Information and Declarations

Competing Interests

Author Contributions

Data Availability

Nigel R. Andrew serves as an Academic Editor for PeerJ.

Behnaz Ghaedi conceived and designed the experiments, performed the experiments, analyzed the data, contributed reagents/materials/analysis tools, wrote the paper, prepared figures and/or tables, reviewed drafts of the paper.

Nigel R. Andrew conceived and designed the experiments, wrote the paper, reviewed drafts of the paper.

The following information was supplied regarding data availability:

Figshare: 10.6084/m9.figshare.2056656.

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
