# Peer review of "The physiological consequences of varied heat exposure events in adult Myzus persicae: a single prolonged exposure compared to repeated shorter exposures"

_PeerJ, doi:10.7717/peerj.2290_

## Round 0.1 · original submission · Major Revisions

Physiological responses of insects to changing environments are an important and often neglected issue. In this manuscript, the authors compare several physiological parameters of aphids exposed to single and repeated heat stress with the control maintained at standard conditions. The research idea is interesting and the approach seems appropriate, however, the manuscript needs a major revision.

First, the paper should be revised with greater emphasis on spelling and grammar.

Also, as pointed out by the Reviewers, the introduction and the discussion are repetitive and should be re-written, including only relevant information. There is too much focus on Australia and aphids' diet and too little on the novelty of the work.

The authors should include all the information missing from the Materials and Methods section, as well as address all the comments concerning bibliography (including literature on symbionts of aphids) and mis-usage of scientific terminology.

All bar plots should be replaced with dot plots, which would immediately make the results more transparent. As it is, for figure 4 it is not clear if four groups of two aphids gave four or eight experimental measurements. This would also help assessing the variability between measurements and the magnitudes of the differences claimed by the authors.

Replication is another important issue raised by Reviewer 1 and it is certainly connected to experimental variability and homogeneity of the genetic background of the animals used in the study (raised by Reviewer 3). The thermal limits experiment is certainly well replicated (3 x 40 aphids), however, 2x4 aphids used for metabolic measurements seem little, especially if there is genetic variability between the animals. I could not find the information on the number of replicates in Figure 5. I think it is necessary for the authors to address these comments.

Also, numerous references to aphid fitness (cost and benefit, fitness in the light of climate change) should be either removed or supported by survival curves. If the authors decide to remove them, the discussion should include the remark made by Reviewer 3: the changes in metabolite content could be due to the aphids' bad health condition and not to adaptation.

I have two additional minor comments:

- red dots indicating sampling points are missing from the control line, Fig. 2.
- the Conclusions section does not include conclusions from this study but authors' recommendation for the scientific community. This section should be removed, filled with conclusions from the study presented in the manuscript or re-named to Perspectives.

·

Basic reporting

This paper covers the interesting topic of repeated heat shock in a M. persicae. While this is a worthwhile field of study, and is highly relevant with the current trend moving away from studying insects at mean temperatures, the paper is poorly written and requires a thorough rewrite.

The relevance of numerous paragraphs is unclear and the paper would benefit from a clearer structure. In addition, some sentences are incomplete, and there are numerous spelling and grammatical errors.

The introduction gives the background context for the study, but the relevance of using this species needs to be highlighted. The literature has not been well cited and at times, more relevant citations are needed. I have listed the references that I noticed were clearly incorrect in the general comments, but I didn’t check each and every one. I encourage the authors to look at the selection of their references carefully. One notably absent reference was the recent review: Colinet et al., 2015, Ann Rev Entomology. Insects in fluctuating thermal environments.

The authors need to revisit the literature w.r.t. the use of fluctuating vs repeated stress terminology. I don’t think that this experiment is a fluctuating thermal regime, it’s more of a repeated heat exposure (or even heat shock/hardening) regime. The terminology of CTmax and lethal temperatures also do not match thermal tolerance norms. Clearing up the terminology will improve the clarity of the paper.

The figures are clearly presented. However, they are filled with numerous small errors – list under general comments. The figure legends could also be revised for clarity.

Experimental design

This original study is well designed but the lack of clear methodological details for experimental conditions, CTmax determination, metabolic reserves and metabolic rate measurements make it impossible to determine the scientific soundness of the research. In addition, the key requirement of enabling others to reproduce the research is not currently met.

The knowledge gap that this study fills could be clearer in the introduction, with some of the necessary information currently buried in the discussion.

Validity of the findings

As stated above, the validity of the findings is difficult to assess given that key information is missing from the methods. I worry about the low replication for these plate-based nutritional assays. These are usually highly variable when performed in small quantities, with small biological replication, and this may be confounding the results.

Additional comments

In general, I think that you have a good study here, but the paper isn’t in great condition which makes it hard to assess the validity of the findings. A thorough rewrite will be interesting to read. I have tried to be as thorough as possible with these comments to guide the improvement of the paper.

The introduction and discussion specifically require a major writing overhaul. The basic structure and flow of logic is missing, and the paper appears to reflect a collection of somewhat connected concepts rather than a focused scientific study. The discussion specifically is currently difficult for me to follow. The authors need to take care to avoid incomplete sentences, spelling and formatting errors. The sheer number of them in this paper is distracting to the reader and I’ve only listed some below. I have focused my specific comments below on the introduction and methods sections. As stated above, the results appear to be sound, but that cannot be evaluated without clear methodological details. The discussion largely just needs to be shorter and more focused.

Introduction

The introduction needs to be clearer on what’s known about this species (not just aphids in general). For example, what temperatures routinely experienced by M. persicae in the field (and what microhabitats do they occupy)? What is the life history of this aphid and are they seasonal pests? What thermal parameters are currently known? Very importantly, do aphids have a heat hardening response?

L49 – Correct “may reduce” to “may be reduced”
L60 – Delete spare )
L72 – Strange sentence: “…if they are not able to move and geographical regions which enable them…”
L76 – Is there a seasonality to aphids as pests?
L76 – All aphids, or specific species
L80 – Is 38.5degC valid for all aphid species? Surely more recent papers have investigated this more specifically since 1951
L82 – What about daily fluctuations?
L86 – This use of aphids requires further justification. This is currently insufficiently motivated in the paper.
L95 – It’s unclear why there is a paragraph on the effects of diet? I think you want to justify your use of artificial diet? Have differences in thermal tolerance for natural vs artificial diet for this species been shown?
L115 – Be clear on the terminology for heat tolerance. Define CTmax. Note that this is not upper lethal temperature. “Critical temperature threshold (L144).
L116 – Karley et al., 2002 – Incorrect citation. This paper does not cover CTmax.
L135 – There is newer literature on the benefits of studying temperature variation and not the mean. Fischer et al., 2011, Physiological Ecology, comes to mind but there are others.
L149 – Terblanche et al., 2010 – Incorrect citation. This paper does not cover recover/repair.

Methods

L158 – “mean temperatue of 25degC” – what was the actual temperature variation
L167 – What is the relevance of this paragraph on diet cages and feeding?
L174 – The diet information could be made supplementary.
L186 – Teet et al. 2010 – incorrect citation. Should be Teets et al., 2011. I don’t see the relevance of this citation. Firstly, it’s a repeated cold experiment, and secondly it was cycles of 12h -5degC, 12h 4degC. This is not even close to your design of 23h 25degC, 1h 38degC.
L187 – How old were the adults? Were they feeding continuously during the experiments?
L187 – Was relative humidity controlled during the 38degC exposure?
L184 – For all experimental conditions, please state whether insects were transferred directly between temperatures, or whether ramping protocols were used. Justify the reasons for your choice. This will affect the interpretation of your results into an environmentally relevant context.
L195 – Be careful with the terminology: “calculated lethal temperature (CTmax)” – CTmax is a performance limit, not a measure of lethal temperature limits. Also an incorrect citation of Davis et al., 2006 – they did not measure CTmax.
L204 – Typo – “determin”
L204 – Justify the use of two different ramping rates during heating. How does this affect the results?
L208 – Describe exactly how the cessation of movement was determined? What thresholds were measured? This needs to be describe in sufficient detail to enable replication
L209 – font change
L209 – Another terminology problem. The measurement you’ve described, is of CTmax or heat coma, and not upper thermal tolerance.
L210 – Were aphids sexed?
L212 – This section on quantification of metabolic reserves needs additional information: full kit details (product numbers), references, number of groups per treatment (biological replicates), correct units (“mg protein ml21”?), spectrophotometer details (make, model). Ridley et al., 2012 does not describe the smaller volumes.
L234 – Error in reporting: “…connected to a Sable Systems International (SSI) (Las Vegas, USA)” – clearly some information is missing
L233 – subscript needed for CO2
L239 – space missing
L241 – How many aphids were in the cuvette?
L244 – Describe the temperature regime. The data does not show any metabolic rate data at different temperatures.
L246 – How long was each aphid recorded for? I see later (L259) is was 2.3h. This is a long time for small insects to spend in dry air. Did the desiccating affect the metabolic rate measurements?
L247 – Activity cannot be recorded with an AD-1 itself. Clarify the methodology.
L248 – This is standard practice. I’m not sure that it requires a citation.

Results

L267 – It would be nice if the order of the results matched the methods section.
L270 – How was dry mass obtained? This detail is missing from the methods section
Figures: All y-axis should have titles and units, not one or the other
Fig 2 – Show sampling timepoints for C1-C3 referred to in Fig 3
Fig 4 and 5 have spelling errors on y-axis: Teatments and Treatmenst.
Fig 4 and 5 – Why were different timepoints used for these parameters? This information should be in the methods section. Is C in Fig 4 from C1, C2 or C3? Same goes for M. I have the same issue with the data presentation in Table 2? I assume the comparisons were made at C3 and M3, but a reader shouldn’t have to make these assumptions.
Fig 5B – “lose water” should be water loss rate

Reviewer 2 ·

Basic reporting

The writing of the manuscript significantly detracts from the overall reading. Extensive grammatical errors, spelling mistakes, and confusing sentence structure overall hinder my enthusiasm for the manuscript. Both the introduction and conclusion are not well organized and overly long, and include numerous digressions into irrelevant details or previous research. As one example, the discussion of the conservation of heat resistance (lines 363-372) bears little relevance to the current work unless the authors are examining this trait across genotypes or species.

The manuscript fails to mention the substantial work done on heat tolerance in other aphid species, notably Acyrthosiphon pisum. Much of this work investigates the role of symbiotic bacteria in aphid heat tolerance, which the authors did not examine. While the aphids’ response to heat is worthy of investigation regardless of the presence or absence of symbionts, knowing which secondary symbionts the aphids are or are not infected with is an important factor when considering the broader claims made regarding aphid tolerance to heat. Similarly, the primary endosymbionts of aphids (Buchnera aphidicola) are highly susceptible to heat stress, and likely have a large impact on the tolerance or lack thereof in Myzus. This literature is fundamental to the project at hand and deserves to be examined in the context of the current work. Some citations to consider:

Burke G, Fiehn O, Moran N. 2009. Effects of facultative symbionts and heat stress on the metabolome of pea aphids. The ISME Journal 4: 242-252. doi:10.1038/ismej.2009.114

Burke GR, McLaughlin HJ, Simon JC, Moran NA. 2010. Dynamics of a recurrent Buchnera mutation that affects thermal tolerance of pea aphid hosts. Genetics 186(1): 367-72. doi:10.1534/genetics.110.117440

The discussion of symbionts in the conclusion (lines 401-407) misrepresents the findings of Dunbar et al (2007) and also seems to suggest that the role of primary endosymbionts in providing amino acids to their hosts is “putative.”

Experimental design

Experimental design
Did the authors establish their aphid lines from a single viviparous female? One of the greatest strengths of research performed on aphids is that their genotype can be completely controlled for, yet the authors do not seem to have controlled for this (lines 154-158). The maternal/grand-paternal effects the authors control for only eliminate environmentally induced variation in aphid nutritional status or stress. Given the possibility of genetic variation within the M. persicae colony used, along with the well-established importance of variation in symbionts, it seems that this is an essential experimental issue to consider. If the aphid genotype was controlled for, it should be explicitly stated, as all interpretation of the results is contingent upon a uniform genetic background.

Were the aphids fed during the course of the experiments? This is unclear from the methods. If the artificial diet were exposed to high temperatures, it would likely change the nutritional value of the diet relative to the control diets that were not heated. Alternatively if the aphids were not fed during the course of the experiment, it would be difficult to differentiate between a nutritional and thermal stress responses.

Validity of the findings

The authors state that the number of high temperature events is more important than the duration of events as determined by the changes in metabolites observed in the repeatedly heated treatments (line 444-446). The authors didn’t test this – the relevant comparison would be the same number of treatments at higher and lower elevated temperature at for two durations.

In general, any mention of the relative effects (better/worse, improvement/decline, etc) of heat stress duration should be limited to discussions of the changes in metabolites. The authors did not assess fitness, and so it is impossible to determine how these changes in physiology will impact aphid fitness.

Additional comments

Ghaedi and Andrew present findings on the heat tolerance of Myzus persicae under constant or fluctuating heat regimes. The authors examined several physiological parameters in response to the different heat stress regimes, including trehalose, glucose, triglyceride, glycerol, and protein content. Their findings indicate that protein and glycerol levels increase while trehalose and triglyceride levels decrease in repeated heat exposure relative to aphids under a constant heat stress. The authors posit that these experiments more accurately represent heat stress conditions faced by insects in the wild, and that the changes in metabolite levels are stress responses.

Reviewer 3 ·

Basic reporting

No Comments

Experimental design

See my comment in the letter to authors on measuring aphid fitness (development and survival)

Validity of the findings

The authors made a good job in putting their results relative to other studies done with the same insect species. The study is, however, somehow limited in showing how the results are novel for the whole community of entomologists, or even for those working with aphids. As mentioned in my comments for the author, the variables measured are convincing, but it is difficult to reconcile them without measures on the effect of the treatments on insect fitness.

Additional comments

In this study Ghaedi and Andrew study physiological responses of the aphid M. persicae to increased temperatures at either constant or fluctuating regime. The authors measure several different molecules in the aphids after experimentally exposing them to such treatments. The authors frame their study in view of global changes including global warming and increase prevalence of heat shocks. The topic is timely and relevant for the readers of a multidisciplinary journal as PeerJ. The paper is well written although, to my taste, a bit too long and repetitive at certain points. My main concern is that the authors argue that their study is relevant in view of the global changes that our planet is experiencing, as it can explain insect resistance or susceptibility to these changes. I agree with that, but I find a bit difficult to interpret the results if aphid fitness is not measured at all. Aphids are exposed to quite dramatic increases in temperature and this (as expected) leads to physiological changes. However, are the authors measuring a real effect on aphid physiology, or just temperature-mediated changes in some molecules in a dying (or already dead) aphid? If this question is not resolved, as I said, the interpretation of the results is quite complicated. A simple survival and development time assay would solve this problem.
I am also a bit concerned about the length of the introduction and the discussion sections. Since the discussion is a bit too speculative as aphid fitness is not measured, I will limit my comments to the introduction section, which I found a bit too long and not very well structured. The authors mention many different topics, but it wasn't clear to me which ones were the most relevant for the results presented. I'd suggest to make it much shorter, for example, in the first 5 paragraphs the authors talk about the effect of temperature, global change and extreme events on insects. I do believe the information presented could be compressed into 2 paragraphs. The same applies when introducing aphids as their model system. In L104-118, many molecular and physiological responses of insects to thermal stress are introduced, are they all investigated in this study? Or the authors just aim at presenting a comprehensive view of the topic. If so, this paragraph is too detailed. If the idea is to introduce some variables that were measured, some information can be placed in the methods section. L120-135, this paragraph would read better if included earlier when global change and heat shocks are introduced.

Minor comments:
The first sentence of the abstract could be a bit more general and not just based on the particular aphid species.
L35-42. Why the introduction is so centred on Australia? I see the authors are from there, but the readers always appreciate a more general view.
L47-51. This paragraph reads a bit confusing.
L54-56. This is not true, non ectoterms are also affected by temperature.
L76. I don't think aphids are pests particularly on canola. There are other cops where aphids too cause serious problems.
L84. what is meant by "interrupt trophic cascades" ?
L95-102. This is mainly a methodological note.
L160. I'd mention parthenogenetic, rather than telescopic reproduction

---

## Round 0.2 · Minor Revisions

Thank you very much for the introduced changes, as they greatly improved the manuscript. I am pleased to inform that your manuscript will be accepted for publication as soon as you address to the satisfactory level the concerns raised in the previous round of revision or arising from it:

1. The translation of the metabolic changes to aphid fitness is not addressed experimentally. I would strongly recommend to discuss this issue, as suggested by the Reviewer.

2. Thank you very much for an attempt of a graph correction. However, these are not dot plots, but bar plots (with a bar replaced by a dot). A dot plot shows each item of numerical data, so one dot for each measurement, and not for each experimental treatment.

Figure 2: Prolong and Repeat treatment lines overlap, which is misleading because the dot on the repeat measurement seems to be common for the prolonged one.

Also, just a curiosity question: why is M an abbreviation for repeated regime? Wouldn't R be more suitable?

4. Please, carefully revise the text, as there are still many grammar mistakes. I listed few below.
Line 79-81 - grammar problem
line 110-111 - studying performance
thermal response
line 118-119 - fitness could be more significant in
repeated temperatures
line 140 - The lab colony were established as an isofemale line
line 144 - Aphids have parthenogenetic
line 169 - diets were changed twice weekly

Reviewer 3 ·

Basic reporting

After the revision the manuscript has greatly increased in clarity, especially the introduction and the discussion sections. This is satisfying, in my previous revision I mentioned this:
"My main concern is that the authors argue that their study is relevant in view of the global changes that our planet is experiencing, as it can explain insect resistance or susceptibility to these changes. I agree with that, but I find a bit difficult to interpret the results if aphid fitness is not measured at all. Aphids are exposed to quite dramatic increases in temperature and this (as expected) leads to physiological changes. However, are the authors measuring a real effect on aphid physiology, or just temperature-mediated changes in some molecules in a dying (or already dead) aphid? If this question is not resolved, as I said, the interpretation of the results is quite complicated. A simple survival and development time assay would solve this problem."

This question hasn't been solved at all, and the authors haven't replied to it properly in their letter. I understand the authors may not be able to add extra experiments, but at least should acknowledge this limitation in the discussion section.

Experimental design

No comments

Validity of the findings

No comments

Additional comments

See above

---

## Round 0.3 · Minor Revisions

Thank you very much for your reply and an attempt of manuscript correction. Unfortunately, many of the problems raised in the previous rounds of revision persist, and I have to ask you for a more thoughtful revision.

1. The authors agreed with the reviewers (in the first round of revision) that the translation of the metabolic changes to aphid fitness is not addressed experimentally, and all conclusions about aphids fitness should be removed. Nevertheless two paragraphs of the discussion still imply that authors measured aphid fitness. Please note that cost and benefit of metabolic changes clearly refer to fitness cost and benefit, despite the omission of the word “fitness”. The two paragraphs are the following:

Paragraph starting at line 378: Our results indicate that repeated high temperature exposure is costly for aphids compared toprolonged high temperature exposure, as there is a significantly lower amount of trehalose and triglyceride production after repeated high temperatures.

Paragraph starting at line 318: We observed increased costs and benefits during repeated heating exposure: - cost and benefit refer to fitness and authors agreed to remove references to fitness.

2. Authors should note that their paper still lacks the exact information about sample sizes and experimental variability between samples. I consider this important, as one of the reviewers in the first round of revision has raised an issue of small sample size and large variability in the metabolic measurements. If the change of the plot format to more informative (dot plot) is impossible I suggest the following:

a) The inclusion of the information on the sample size for each figure, starting with figure 4. The description should be analogous to the one already present for figure 3 legend (“The metabolite content is based on the mean of four replicates, two aphids per replicate for the all metabolites”).

b) In table 1, next to the mean value for each measurement, I would suggest to include the column with all the values obtained for each measurement.

c) Remove "Dot plot" from the figure legends.

3. I would like to ask for the third time for a careful revision of the text, as there are still many mistakes, for example:

Line 191 - For the single prolonged heat exposure treatment, separate groups of adults were exposed to 38°C for 3h, because after three cycles of repeated heat stress, adults have accumulated 3h at 38°C. - what have they accumulated?

Line 200 - because as their – please, choose one, “as” or “because”

Line 211 - The assay kits :used – Please, check the placement of all colons and semicolons in the text.

Line 217 - and following trehalose measurement (Porcine Kidney Trehalase (Sigma:T8778) 1 unit per mg of protein, 3.7 mg per ml and 0.2 M sodium citrate (5.882 g/100ml), 1 mM Sodium EDTA (37.22 mg/100 ml), D+Trehalose dihydrate (Sigma:T0167)). - The meaning of this sentence, from 1unit per mg of protein is unclear.

Line 262 - at which ceased walking and succumbed to uncontrollable spasm (Hazell et al., 2008; Alford et al., 2012b). - should be: “walking ceased and aphids succumbed”

Lines 300-302 - triglyceride content was less than half that of control (0.659±0.01) and prolonged (0.597±0.01) aphids and reached 0.324±0.02 μg triglyceride mg-1 (P<0.002).(Fig. 3E). - I am sure authors do not mean “prolonged aphids”.

Line 316 - consequences – should be “consequences of”

---

## Round 0.4 · Minor Revisions

Thank you very much for the introduced changes, as I think they have improved the manuscript so far. However, I still have to ask for the proper presentation of the data. I also cannot agree with the argument the authors have raised before: it has been published this way before so it must be right and correct. I believe PeerJ has high standards on realistic and adequate data presentation, probably higher than other journals, and the variability in the data presented by the authors is impossible to asses. I recommend a great review by Motulsky discussing the use of mean and SEM for the data presentation (http://jpet.aspetjournals.org/content/351/1/200.long) and I recommend the detailed inspection of the figure 5 (the same dataset presented in many ways). As the authors decided that the graph format cannot be changed, I am repeating the request for the inclusion of the raw data.

---

## Round 0.5 · accepted · Accept

Thank you very much for the link to the data. I believe your manuscript now fulfils all the requirements for publication at PeerJ. Congratulations and good luck with the future scientific endeavours!